# Epidemiology of West Nile virus in Africa: An underestimated threat

Giulia Mencattelli[1,2], Marie Henriette Dior Ndione[3], Roberto Rosà[1,2], Giovanni Marini[1], Cheikh Tidiane Diagne[3], Moussa Moise Diagne[3], Gamou Fall[3], Ousmane Faye[3], Mawlouth Diallo[4], Oumar Faye[3], Giovanni Savini[5], Annapaola Rizzoli[1]*

**1** Department of Biodiversity and Molecular Ecology, Research and Innovation Centre, Fondazione Edmund Mach, San Michele all'Adige, Trento, Italy, **2** Center Agriculture Food Environment, University of Trento, San Michele all'Adige, Trento, Italy, **3** Department of Virology, Fondation Institut Pasteur de Dakar, Dakar, Senegal, **4** Department of Zoology, Fondation Institut Pasteur de Dakar, Dakar, Senegal, **5** Department of Public Health, OIE Reference Laboratory for WND, Istituto Zooprofilattico Sperimentale dell'Abruzzo e del Molise "G. Caporale", Teramo, Italy

* annapaola.rizzoli@fmach.it

**Data Availability Statement:** All relevant data are within the manuscript and its Supporting Information files.

**Funding:** The authors received no specific funding for this work. This study was partially funded by

## Abstract

### Background

West Nile virus is a mosquito-borne flavivirus which has been posing continuous challenges to public health worldwide due to the identification of new lineages and clades and its ability to invade and establish in an increasing number of countries. Its current distribution, genetic variability, ecology, and epidemiological pattern in the African continent are only partially known despite the general consensus on the urgency to obtain such information for quantifying the actual disease burden in Africa other than to predict future threats at global scale.

### Methodology and principal findings

References were searched in PubMed and Google Scholar electronic databases on January 21, 2020, using selected keywords, without language and date restriction. Additional manual searches of reference list were carried out. Further references have been later added accordingly to experts' opinion. We included 153 scientific papers published between 1940 and 2021.

This review highlights: (i) the co-circulation of WNV-lineages 1, 2, and 8 in the African continent; (ii) the presence of diverse WNV competent vectors in Africa, mainly belonging to the *Culex* genus; (iii) the lack of vector competence studies for several other mosquito species found naturally infected with WNV in Africa; (iv) the need of more competence studies to be addressed on ticks; (iv) evidence of circulation of WNV among humans, animals and vectors in at least 28 Countries; (v) the lack of knowledge on the epidemiological situation of WNV for 19 Countries and (vii) the importance of carrying out specific serological surveys in order to avoid possible bias on WNV circulation in Africa.

### Conclusions

This study provides the state of art on WNV investigation carried out in Africa, highlighting several knowledge gaps regarding i) the current WNV distribution and genetic diversity, ii)

the EU Horizon 2020 Framework Program, grant 874850 MOOD and is catalogued as MOOD 22. This study was supported by an international PhD initiative including Fondazione Edmund Mach, University of Trento, Istituto Zooprofilattico of Teramo, and Institut Pasteur of Dakar. The funders had no role in study design, data collection and analysis, decision to publish, or preparation of the manuscript.

**Competing interests:** The authors have declared that no competing interests exist.

its ecology and transmission chains including the role of different arthropods and vertebrate species as competent reservoirs, and iii) the real disease burden for humans and animals. This review highlights the needs for further research and coordinated surveillance efforts on WNV in Africa.

## Author summary

Since its discovery in the African continent in 1937, West Nile virus expansion and invasion into new regions represent a serious concern today for an increasing number of countries worldwide. Although about 80% of infected individuals are asymptomatic, this zoonotic virus is pathogenic for humans other than for some animal species, displaying a range of clinical manifestation spanning from influenza-like symptoms to severe neurological complication and death.

This study provides an updated overview on the current knowledge of WNV epidemiology in each African country, summarizing available data on incidence of the infection in humans and animals, the circulating lineages and clades, other than an updated list of the principal arthropod vectors found naturally infected and the availability of vector competence studies. However, this review highlights also the lack of knowledge regarding the occurrence and intensity of circulation of WNV in many African countries. Therefore, considering the sensitivity of WNV transmission system to climate and other environmental changes, along with the increasing level of interconnections among continents due to globalization, the intensification of the research activities on WNV and a promotion of a coordinated surveillance actions across African and European countries would provide the information of utility for a better evaluation of the actual risk of WNV and disease burden at trans-continental scale.

## Introduction

West Nile virus (WNV) is a mosquito-borne virus, part of the genus *Flavivirus*, family *Flaviviridae* and member of the Japanese Encephalitis virus serocomplex which includes other closely related viruses such as Saint Louis encephalitis, Usutu, Kunjin, Kookaburra, Stratford, Alfuy and Murray Valley encephalitis [1,2]. WNV is endemo-epidemic in Africa, Europe, Middle East, Asia, and the New World, representing an emerging threat for public and animal health due to the continuous expansion of its range [2,3].

The first description of WNV dates back to 1937 when it was reported from Omogo, in the West Nile district of the Northern province of Uganda following a campaign aimed at monitoring the circulation of Yellow Fever virus [4]. The principal vectors of WNV are mosquitoes, mostly belonging to *Culex* spp. and *Aedes* spp. [5]. Other arthropods found naturally infected with WNV are ticks, although their role as competent vector is still not well understood [6,7]. WNV has been identified in several vertebrate species, especially birds belonging to the order Passeriformes [8]. Other species in which WNV has been reported include Piciformes, Columbiformes, Charadriiformes, Falconiformes, Strigiformes, Anseriformes, Psittaciformes, and Galliformes [2]. The role of these species as real competent reservoir hosts has been proved only for a limited subset [7]. Humans, horses, and other vertebrate hosts are considered WNV dead-end hosts, since they are susceptible to the infection but unable to transmit the virus to mosquitoes [7].

WNV infection is mostly asymptomatic but a range of clinical forms and symptoms are reported for humans, horses, and birds [7]. In humans, around 20% of cases develop influenza-like symptoms (West Nile fever, WNF), while less than 1% develop the West Nile Neuroinvasive Disease (WNND) with encephalitis, meningitis, acute flaccid paralysis, and occasionally death [2]. The severity of symptoms generally depends on WNV strains involved other than to the general physical conditions of the patients [5]. In domestic animals, such as horses, only 20% of WNV-infected individuals show mild symptomatic infections while 1–10% are characterized by severe neurological disease with a mortality rate of about 33% [7]. Among birds, corvids and raptors appear highly susceptible to WNV infection, resulting in higher incidence with severe neurological signs that lead the individuals to death [9,10].

WNV currently includes up to nine phylogenetic lineages, identified through phylogenetic analyses: WNV lineage 1 (WNV-L1) to lineage 9 (WNV-L9) [11]. WNV-L7 has been recently classified as a distinct flavivirus, the Koutango virus (KOUTV) [2,12]. Among all these observed lineages, only WNV-L1, L2, and L8, other than the KOUTV, have been detected in Africa [2]. WNV-L1 and L2 are the most important from the public health point of view because they are most pathogenic and widespread, and implicated in several outbreaks worldwide [3,7]. WNV-L1, mainly diffused in Central and Northern Africa, emerged in Europe in the 1960s [3]. After 30 years of silence, it started causing epidemics in North America, Northern African, Western, and Eastern European countries [3]. The main actor of the European scenario was WNV-L1 up to 2004 when WNV-L2, considered endemic in Southern Africa and Madagascar, was reported for the first time in Hungary [3]. Since 2010 it started causing several outbreaks in central Europe and it is nowadays one of the main lineages responsible for WNV infections in Europe [3,7].

Other less widespread lineages are WNV-L3, also known as Rabensburg virus, present in Czech Republic; WNV-L4, isolated and reported in Russia; WNV-L5, isolated in India and often considered as the clade 1c of WNV-L1, and WNV-L6, based on a small gene fragment, isolated in Spain [1,2]. Finally, putative lineage 9, often considered a sub lineage of WNV-L4, has been isolated from *Uranotaenia unguiculata* mosquitoes in Austria [11]. These lineages, never isolated in Africa, might have evolved from distinct introductions into the Northern Hemisphere [13].

Translocation of diverse WNV lineages from the original ecological niches to new geographic areas is generally thought to occur mainly through migratory birds, although the final chain of events that lead to the introduction or reintroduction of the virus into new continents needs further explanations [7]. For example, phylogenetic analyses revealed that all European WNV-L1 and 2 strains are derived from a limited number of initial independent introductions, most likely directly from Africa, followed by local amplification and spread [14,15].

WNV current distribution, genetic variability, ecology, and epidemiological pattern in the African continent are only partially known despite the general consensus on the urgency to obtain such information for quantifying the actual disease burden in Africa other than to predict future threats at continental and global scale.

Therefore, we performed a systematic review with the aim to provide an updated overview of the current knowledge regarding WNV epidemiology in Africa, its major features in terms of geographical distribution, molecular diversity and phylogeography, principal vectors and hosts, human and animal epidemiological patterns.

This information would provide an updated overview and data of utility for better quantifying the actual risk and disease burden in Africa other than predicting future threats at global scale.

## Materials and methods

### Search strategy and selection criteria

Pertinent articles were searched, screened, and incorporated in the Systematic Review according to PRISMA and QUORUM criteria [16]. Relevant background information was obtained by searching on the PubMed and the Google Scholar electronic databases on January 17, 2020 (n = 375), using the search terms "West Nile virus" and "Africa" with no restrictions on the earliest date of the articles returned. Additional records have been identified through contact with experts (n = 33).

Studies were classified by topic (West Nile vector-borne disease) and Continent (Africa). Each search was conducted with common variations of the virus name, specifically: West Nile virus, WNV; and the geographic region intended to be studied, specifically: Africa. Full-text original articles were searched. After removal of duplicates, two reviewers independently screened articles by title and abstract. Finally, pertinent records were selected for full-text screening and, if relevant, included in the review (for details see S1 Fig).

Documents were included if containing the following information: i) general overview of WNV features and distribution; ii) WNV phylogenetics, including a description of the biology, phylogenetic and phylogeography of WNV lineages over all continents but focusing mainly on Africa; iii) WNV main vectors and animal hosts; and iv) human epidemiology, with all the information related to the virus and the human infection along with a detailed report of molecular and serological studies. Two reviewers processed the document evaluation based on articles designed for full-text review.

## Results

### Article's selection process

We identified 408 articles. After duplicates were removed, the remaining 395 records were screened by title, abstract, and full text, resulting in 84 studies which were finally included into the review. Reference lists of the included studies were further screened for relevant research. Following the same eligibility criteria, 69 citations were incorporated in the study. Finally, a total of 153 studies, including 84 full-text reviewed articles and 69 citations were considered.

### 1. Genomics and phylogeography

WNV is a biologically diverse virus, characterized by several genotypic and phenotypic changes [2,7].

Phylogenetic analysis, performed through the construction of evolutionary trees, predicted the time for the WNV most common ancestor (tMRCA) to be between the 16th and the 17th century in Africa [2]. The virus evolution led to the formation of two new branches, one characterized by WNV-L1 and L5, and the other by WNV-L2, L3, L4, L8 and L9 (not enough information is available regarding WNV-L6). WNV-L1 was successfully introduced into Europe in the 1960s while it appeared for the first time in North America in 1999, subsequently becoming endemic across both continents [1]. WNV-L2, after its first appearance in Hungary in 2004, showed multiple introductions into Europe [1,2,17].

WNV-L1 is widespread and frequently associated with symptomatic infections in humans and horses [3]. It includes 3 clades (A, B, and C) and several sub-clades [1,2,15,18]. Only Clade A is widespread in the African continent. Clade A is composed by 6 sub-clusters [14]. Among them, the sub-clusters 1, 2, 3, 5 and 6 have been detected in different areas of the Continent [1]. WNV-L1-Clade A strains belonging to the diverse sub-clusters are all phylogenetically very similar to each other, suggesting local and long-range WNV circulation probably through

migratory birds [19]. Following the introduction of WNV-L1 in the New World, a huge number of sequences have been obtained overtime [3,14,20]. Interestingly, phylogenetic analysis demonstrated that the strain *PaH001* isolated in Tunisia in 1997 roots the tree of WNV-L1 circulating in North America [14,21]. Furthermore, phylogenetic and genetic distance studies evidenced that the Tunisian strain *PaH001* is closely related to a group of highly conserved viruses collected in America and Israel between 1998 and 2000, suggesting that viruses circulating in the Middle East / North Africa are related to those circulating in North America [14]. A possible introduction of WNV-L1 in Europe from Morocco is also suspected: the closest ancestor of the European strains could be a Moroccan strain which appears to be closely genetically related to French and Italian isolates (France: 2000, 2006; Italy: 1998, 2008) [3]. Up to date, WNV-L1 has been reported in the following African countries: Algeria, Central African Republic, Egypt, Côte d'Ivoire, Kenya, Morocco, Tunisia, Senegal, and South Africa [1,2,14,15,18,19,21–23].

WNV-L2 was considered for a long time to be less pathogenic than WNV-L1, until it evolved [six amino acid substitutions at the level of the E (V159I), NS1 (L338T), NS2A (A126S), NS3 (N421S), NS4B (L20P) and NS5 (Y254F) proteins] becoming more virulent and causing also severe disease forms in South Africa other than among humans and birds in Europe [2,17,18]. It is now endemic and it is the most prevalent lineage circulating in several African and European countries [18,24].

In Africa, WNV-L2 circulation has been reported in Botswana, Central African Republic, Congo, Djibouti, Madagascar, Mozambique, Namibia, Senegal, South Africa, Tanzania, and Uganda [1,2,13,17–19,22,23,25–27].

WNV-L2 includes 4 clades (A-D) all circulating in Africa [17,18,27]. Clade A is characterized by strains circulating in Madagascar, Senegal, and Uganda. Clade B is composed of three main subclades: Cluster 1, 2 and 3. Among them only cluster 3 occurs in Africa (Madagascar, Namibia, and South Africa) [17,27]. Clade C is characterized by strains circulating in Madagascar while Clade D, the most widespread, is composed of strains circulating in Central African Republic, Congo, Namibia, Senegal, South Africa, and Uganda [17,27].

The exact origin of WNV-L2 strains and the following route of introduction into Europe are not clear [3,28]. Detected for the first time in Hungary in 2004, WNV-L2 then spread into many European wetland areas, such as the Aliakmonas Delta in northern Greece (2010), and the Po Delta in north-east Italy (2011) [28]. These areas are along the major flyways of birds migrating from Africa, thus supporting the hypothesis of a possible role of migratory birds for the introduction of WNV from African countries into Europe [12,28].

Additional lineages were discovered in Africa as WNV-L7 (KOUTV) and putative Lineage 8 [12,22].

WNV-L7, reported for many years as a separate lineage of WNV [14], has been recently classified as the KOUTV, which is now considered a distinct flavivirus (https://talk.ictvonline. org/ictv-reports/ictv_online_report/positive-sense-rna-viruses/w/flaviviridae/360/genus-flavivirus). KOUTV, discovered in the Koutango district of the Kaolack region of Senegal, has been isolated from ticks, rodents, and sandflies [12]. Important studies showed a high virulence of Koutango virus in mice and a potential risk for humans has been highlighted following a severe accidental infection in a Senegalese lab worker [2,22]. KOUTV is exclusively present in Africa, circulating in Senegal, Gabon, Somalia, and Niger [12]. Its invasion into other Continents could represent a possible future threat worldwide.

Putative lineage 8 has been isolated from *Culex perfuscus* mosquitoes in 1992 in the Kedougou region of Senegal [22]. It is characterized by low virulence, and this might represent a good feature for making the lineage a good candidate for a new WNV vaccine [2]. Fig 1 shows the currently known WNV lineages reported for 17 countries in Africa.

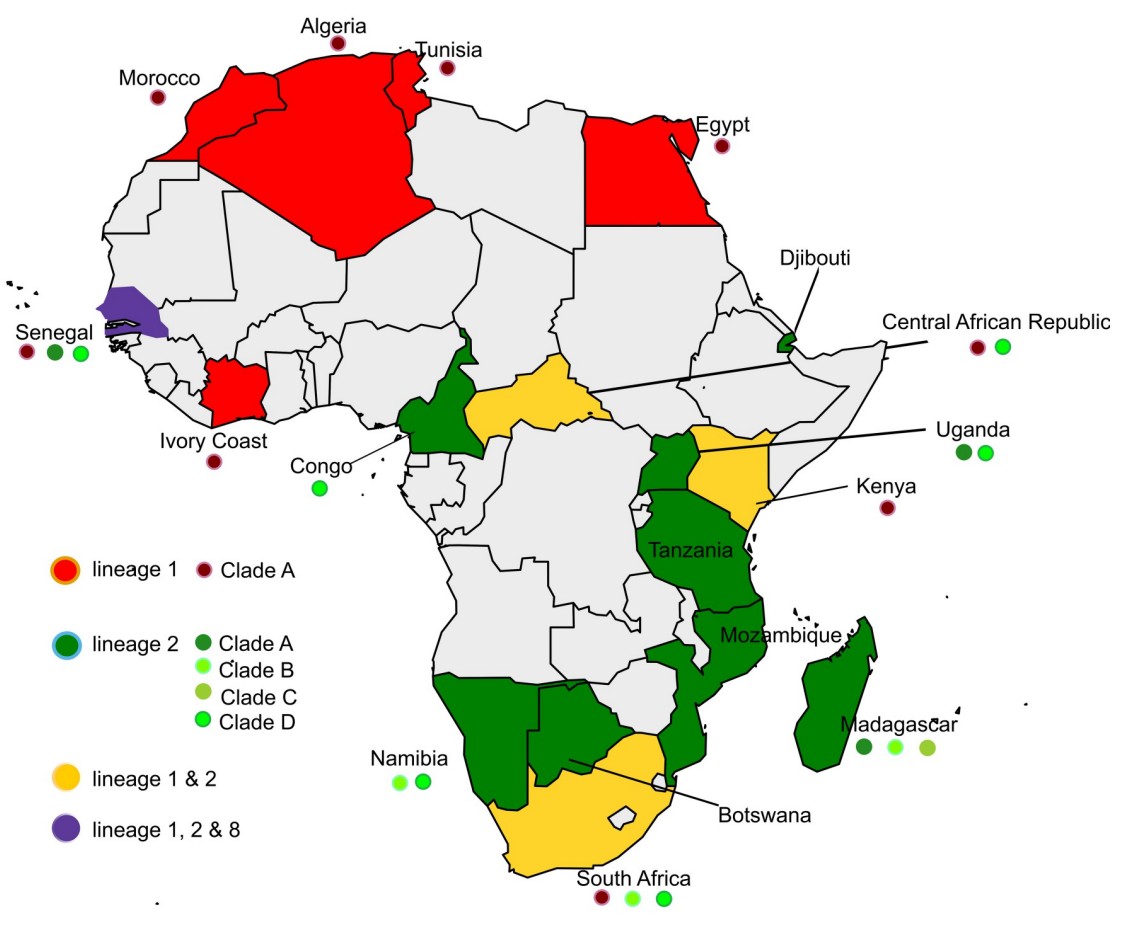

**Fig 1. West Nile virus lineages reported for 17 African countries.** Map was generated using publicly available shapefiles, https://smart.servier.com/category/general-items/world-maps/.

## 2. West Nile virus vectors in Africa

In Africa, the most widespread WNV vectors mainly belong to the *Culex pipiens* complex (*Cx. pipiens* s.l.) [7,29–31] which includes *Cx. p. pipiens* Linnaeus, *Cx. p. pipiens biotype molestus* Forskal, *Cx. p. quinquefasciatus* Say, *Cx. p. pallens* and *Cx. torrentium* Martini [32,33]. As summarized in Table 1, WNV has been isolated from at least 46 mosquito species but studies to assess their vectorial competence have been carried out, between 1972 and 2016, only for 8 mosquito species (*Cx. quinquefasciatus*, *Cx. univittatus*, *Cx. vansomereni Edwards*, *Ma. uniformis*, *Ma. Africana*, *Cx. pipiens*, *Cx. theileri*, and *Cx. neavei*) in Madagascar, the Maghreb region (Algeria, Morocco, Tunisia), Senegal, Kenya, and South Africa (Table 2) [19,22,23,26,27,29,34–50]. These studies provide evidence of the vector competence of *Cx. pipiens*, *Cx. quinquefasciatus*, *Cx. vansomereni*, *Cx. univittatus*, *Cx. theileri*, and *Cx. neavei* mosquitoes in Africa [22,29,33–35,48,51].

**Table 1. West Nile virus isolation and availability of vector competence studies for different mosquito species in Africa.**

| Species | Country | Virus isolation | Vector competence experiment | References |
|---|---|---|---|---|
| *Anopheles brunnipes* | Madagascar | + | - | [36[c],51[c]] |
| *Anopheles coustani* | Madagascar | + | - | [56[b]] |
| *Anopheles maculipalpis* | Madagascar | + | - | [36[c],51[c]] |
| *Anopheles pauliani* | Madagascar | + | - | [56[b]] |
| *Anopheles scotti* | Madagascar | + | - | [36[c],51[c]] |
| *Anopheles* spp. | Kenya | + | - | [37[b]] |
| *Aedeomyia africana* | Senegal | + | - | [36[c],38[a]] |
| *Aedeomyia madagascarica* | Madagascar | + | - | [56[b]] |
| *Aedes albocephalus* | Madagascar | + | - | [36[c]] |
| *Aedes (Aedimorphus) dalzieli* | Madagascar | + | - | [56[b]] |
| *Aedes juppi + caballus* | South Africa | + | - | [34[c]] |
| *Aedes vexans* | Senegal | + | - | [36[c],39[a]] |
| *Aedes madagascariensis* | Madagascar | + | - | [36[c],56[b]] |
| *Aedes albothorax* | Kenya | + | - | [36[c]] |
| *Aedes circumluteolus* | Madagascar, South Africa | + | - | [34[c],36[c],56[b]] |
| *Aedes aegypti* | Madagascar | + | - | [12,35[a],36[c],51[c],56[b]] |
| *Aedes aegypti* | Senegal | - | + (KOUTV) | [12,54,55] |
| *Aedes africanus* | Central African Republic | + | - | [36[c]] |
| *Aedes* spp. | Kenya | + | - | [37[b]] |
| *Aedes* spp. | Senegal | + | - | [12[c]] |
| *Culex antennatus* | Egypt, Madagascar, Senegal | + | - | [36[c],38[a],40[a],48[c]] |
| *Culex decens* group | Madagascar | + | - | [36[c],56[b]] |
| *Culex ethiopicus* | Ethiopia, Senegal | + | - | [36[c],38[a]] |
| *Culex guiarti* | Ivory Coast | + | - | [35[a],36[c]] |
| *Culex neavei* | Senegal, South Africa | + | + | [22[a],34[c],38[a],40[a]] |
| *Culex nigripes* | Central African Republic | + | - | [36[c]] |
| *Culex perfuscus* | Ivory Coast, Central African Republic, Senegal | + | - | [36[c]] |
| *Culex perexiguus* | Algeria | + | - | [23[b]] |
| *Culex pipiens* | South Africa, Egypt, Algeria, Morocco, Tunisia | + | + | [21[b],29,31[c],34[c],36[c],41[b],42[b],45[c],48[c],49[c]] |
| *Culex pipiens* spp. *torridus* | Djibouti | + | - | [26[b]] |
| *Culex poicilipes* | Senegal | + | - | [36[c],38[a],43[a]] |
| *Culex pruina* | Central African Republic | + | - | [36[c]] |
| *Culex quinquefasciatus* | Djibouti, Madagascar | + | + | [26,35[a],36[c],51[a]] |
| *Culex scottii* | Madagascar | + | - | [36[c]] |
| *Culex* spp. | Algeria, Kenya | + | - | [19,37[b],44[a,b]] |
| *Culex theileri* | South Africa | + | - | [34[c]] |
| *Culex tritaeniorhynchus* | Madagascar | + | + | [35[a],36[c],38[a],51[a]] |
| *Culex univittatus* | Madagascar, Egypt, Kenya, Namibia, South Africa | + | + | [27[a,b],34[c],36[c],45[c],46[a],47[a,b],48[c],49[c],51[a]] |
| *Cx. vansomereni Edwards* | Kenya | - | + | [50] |
| *Culex weschei* | Central African Republic | + | - | [36[c]] |
| *Coquillettidia metallica* | Uganda | + | - | [36[c]] |
| *Coquillettidia microannulata* | South Africa | + | - | [34[c],36[c]] |
| *Coquillettidia richiardii* | South Africa | + | - | [34[c],36[c]] |
| *Mansonia africana* | Senegal | + | + | [35[a],38[a]] |

(*Continued*)

**Table 1.** (Continued)

| Species | Country | Virus isolation | Vector competence experiment | References |
|---------|---------|-----------------|------------------------------|------------|
| *Mansonia uniformis* | Senegal, Ethiopia, Madagascar | + | + | [35[a],36[c],38[a],40[a],43 [a],51[a],56[b]] |
| *Mimomyia hispida* | Senegal | + | - | [36[c],38[a]] |
| *Mimomyia lacustris* | Senegal | + | - | [36[c],38[a]] |
| *Mimomyia splendens* | Senegal | + | - | [36[c],38[a]] |
| *Mimomyia* sp. | Senegal, Kenya | + | - | [36[c],38[a]] |

+ At least one study report with positive results found;—no available studies

[a] = "virus isolation on cell cultures / injection into suckling mice. Viruses detected by immunofluorescence assay using specific mouse immune ascitic fluids"

[b] = "RNA molecular detection"

[c] = "viral isolation, not specified if [a] and/or [b]"

no apical letters = studies indicating only vector competent experiments.

*Cx. pipiens*, characterized by high dissemination and transmission rates, are indicated by numerous reports as the most important vector species of WNV [29,31,34,36,41,42,45,48,49].

High ornithophilic and low anthropophilic *Cx. univittatus* mosquitoes are considered both highly susceptible and efficient transmitters of WNV [34]. WNV vertical transmission has been described under field condition in *Cx. univittatus* males in the Rift Valley province of Kenya [38].

*Cx. theileri* is a less efficient WNV vector: despite being highly susceptible to the virus it has a lower transmission rate [34,52]. Vector competent experiments highlight that this mosquito species can be infected as much as *Cx. univittatus*, but probably due to its host preferences (less ornithophilic than *Cx. univittatus* and feeding mostly at ground level) there are very few WNV isolates obtained from wild population, classifying this species as a poor vector [52].

*Cx. quinquefasciatus*, shown to be a competent species for WNV-L1 and -L8, is widespread in urban environments and active all year-round, and might be considered as another important WNV vector, especially in urban settings [22]. Furthermore, vector competence studies highlighted that *Cx. quinquefasciatus* mosquitoes were not competent for KOUTV [22].

*Cx. neavei*, attracted by both horses and birds, and widespread in different type of lands, might have a possible role as bridge vectors in the sylvatic transmission cycle [22]. This mosquito species has been shown to be more efficient vectors for WNV-L1 than L8, and to be susceptible to WNV-L2 and KOUTV infections [22]. However, WNV transmission has not been observed for WNV-L2 and KOUTV [22]. As shown in Table 2, experimental infections conducted on *Cx. neavei* in South Africa [34,53] showed high transmission rate but a 50% infection threshold, observed after exposing birds with different viraemias to this mosquito species, of 4.4 logs per ml, that was higher than those observed in *Cx. univittatus* mosquitoes (2.1 logs per ml) [53]. Furthermore, the time taken by *Cx. neavei* and *Cx. quinquefasciatus* mosquitoes to develop WNV after infection (extrinsic incubation period, EIP) has been estimated to last 15 days at 27˚C [22]. However, the infective life survival rate has been estimated to be comprised between 0.75 and 0.88 for *Cx. neavei* and between 0.87 and 0.88 for *Cx. quinquefasciatus* mosquitoes [22]. Based on these data, only 1.3% to 10.4% of *Cx. neavei* and 12.59% to 15.45% of *Cx. quinquefasciatus* mosquitoes have been estimated to survive at 15 days post infection by the authors of the study [22]. Despite *Cx. quinquefasciatus* and *Cx. neavei* mosquitoes being widely distributed in the African continent and their proven competence for distinct lineages of WNV, such findings imply a low probability of WNV transmission to new hosts in the African continent and they might indicate a low impact of *Cx. neavei* and *Cx. quinquefasciatus* mosquitoes in WNV circulation in Africa [22]. However, the small number of mosquitoes

**Table 2. Quantitative information related to WNV vector competence studies carried out on different mosquito species in Africa.**

| Country | Days post infection | Infection rate | Transmission rate | Dissemination rate | Species [lineage; inoculated virus titer] | References |
|---|---|---|---|---|---|---|
| Kenya | 7*, 13–14**, 20–21*** | 42% | 17%[a], 100%[b], 83%[c], 86%[d] | 0%*, 100%**, 100%*** | Cx. vansomereni Edwards [L1; 10^5.8–7.2] | [35,50] |
| Kenya | (··) | 50% | (··) | 0%*, 0%**, n.t.*** | Mansonia Africana [L1; 10^5.8–7.2] | [35,50] |
| Kenya | 7*, 13–14**, 20–21*** | 82% | 46%[a], 80%[b], 100%[c], 83%[d] | 28%*, 86%**, 94%*** | Culex quinquefasciatus | [35,50] |
| Kenya | 7*, 13–14**, 20–21*** | 51% | 19%[a], 50%[b], 100%[c], 67%[d] | 33%*, 91%**, 100%*** | Culex univittatus | [35,50] |
| Kenya | 7*, 13–14**, 20–21*** | 43% | (··) | 0%*, 100%**, n.t.*** | Mansonia uniformis | [35,50] |
| Maghreb region (Algeria, Morocco, Tunisia) | 3 | (··) | 5% | (··) | Culex pipiens | [29] |
| Maghreb region (Algeria, Morocco, Tunisia) | 14 | (··) | 40% | 59.1% - 100% | Culex pipiens | [29] |
| Maghreb region (Algeria, Morocco, Tunisia) | 21 | (··) | 80% | (··) | Culex pipiens | [29] |
| Senegal | 4 | 14.28% | (··) | 0% | Culex neavei [L1, Titer 10^6] | [22] |
| Senegal | 8 | 14.28% | (··) | 0% | Culex neavei [L1, Titer 10^6] | [22] |
| Senegal | 12 | 25% | (··) | 0% | Culex neavei [L1, Titer 10^6] | [22] |
| Senegal | 15 | 55% | 83.3% | 54.5% | Culex neavei [L1, Titer 10^6] | [22] |
| Senegal | 4 | 0% | (··) | (··) | Culex neavei [L2, Titer: 10^5] | [22] |
| Senegal | 8 | 0% | (··) | (··) | Culex neavei [L2, Titer: 10^5] | [22] |
| Senegal | 12 | 0% | (··) | (··) | Culex neavei [L2, Titer: 10^5] | [22] |
| Senegal | 15 | 6.67% | (··) | 50% | Culex neavei [L2, Titer: 10^5] | [22] |
| Senegal | 4 | 0% | (··) | (··) | Culex neavei [L8, Titer: 10^5] | [22] |
| Senegal | 8 | 0% | (··) | (··) | Culex neavei [L8, Titer: 10^5] | [22] |
| Senegal | 12 | 0% | (··) | (··) | Culex neavei [L8, Titer: 10^5] | [22] |
| Senegal | 15 | 5.55% | (··) | 100% | Culex neavei [L8, Titer: 10^5] | [22] |
| Senegal | 4 | 25% | (··) | 0% | Culex quinquefasciatus [L1, Titer 10^6] | [22] |
| Senegal | 8 | 25% | (··) | 0% | Culex quinquefasciatus [L1, Titer 10^6] | [22] |
| Senegal | 12 | 25% | (··) | 0% | Culex quinquefasciatus [L1, Titer 10^6] | [22] |
| Senegal | 15 | 75.86% | (··) | 18.18% | Culex quinquefasciatus [L1, Titer 10^6] | [22] |
| Senegal | 4 | 0% | (··) | (··) | Culex quinquefasciatus [L2, Titer 10^5] | [22] |
| Senegal | 8 | 0% | (··) | (··) | Culex quinquefasciatus [L2, Titer 10^5] | [22] |
| Senegal | 12 | 0% | (··) | (··) | Culex quinquefasciatus [L2, Titer 10^5] | [22] |
| Senegal | 15 | 5.26% | (··) | 0% | Culex quinquefasciatus [L2, Titer 10^5] | [22] |
| Senegal | 4 | 0% | (··) | (··) | Culex quinquefasciatus [L8, Titer 10^6] | [22] |
| Senegal | 8 | 0% | (··) | (··) | Culex quinquefasciatus [L8, Titer 10^6] | [22] |
| Senegal | 12 | 0% | (··) | (··) | Culex quinquefasciatus [L8, Titer 10^6] | [22] |

(*Continued*)

**Table 2.** (Continued)

| Country | Days post infection | Infection rate | Transmission rate | Dissemination rate | Species [lineage; inoculated virus titer] | References |
|---------|---------------------|----------------|-------------------|--------------------|--------------------------------------------|------------|
| Senegal | 15 | 0% | (··) | (··) | *Culex quinquefasciatus [L8, Titer 10^6]* | [22] |
| Senegal | 15 | 0% | (··) | (··) | *Culex quinquefasciatus [L8, Titer 10^5]* | [22] |
| South Africa | 21 | 97% | 100% | (··) | *Culex neavei [Titer 10^5.7]* | [34,53] |
| South Africa | 12–28 | 97% | (··) | (··) | *Culex neavei [Titer 10^5.7]* | [34,53] |
| South Africa | 13–29 | 24% | (··) | (··) | *Culex neavei [Titer 10^4.0]* | [34,53] |
| South Africa | 15–18 | 4% | (··) | (··) | *Culex neavei [Titer 10^3.7]* | [34,53] |
| South Africa | 15–18 | 8% | (··) | (··) | *Culex neavei [Titer 10^3.2]* | [34,53] |
| South Africa | 17 | 100% | 97% | (··) | *Culex univittatus [Titer 10^5.8–6.3]* | [34,53] |
| South Africa | (··) | 100% | 33% | (··) | *Culex univittatus [Titer 10^4.3]* | [34,53] |
| South Africa | (··) | 84% | (··) | (··) | *Culex univittatus [Titer 10^2.7]* | [34,53] |
| South Africa | (··) | 41% | (··) | (··) | *Culex univittatus[Titer 10^1.9]* | [34,53] |
| South Africa | (··) | (··) | 0% | (··) | *Culex theileri [Titer 10^6.2]* | [34] |
| South Africa | (··) | (··) | 25% | (··) | *Culex theileri [Titer 10^7.1]* | [34] |
| South Africa | 21–22 | 100% | 0%* | (··) | *Culex theileri [Titer 10^4.5]* | [34,52] |
| South Africa | 21–22 | 92% | 0%^ | (··) | *Culex theileri [Titer 10^3.5]* | [34,52] |
| South Africa | 21–22 | 52% | 0%^ | (··) | *Culex theileri [Titer 10^2.5]* | [34,52] |
| South Africa | 21–22 | 14% | 0%^ | (··) | *Culex theileri [Titer 10^1.5]* | [34,52] |
| South Africa | 18 | 100% | 25%^^ | (··) | *Culex theileri [Titer 10^5.4]* | [34,52] |

Infection rate (number of infected mosquito bodies per 100 mosquitoes tested); Dissemination rate (number of mosquitoes with infected legs/wings per 100 mosquitoes infected); Transmission rate (number of mosquitoes with infected saliva per 100 mosquitoes with infected legs/wings); Transmission rate[a]: % of orally exposed mosquitoes (regardless of their infection status) that took a second bloodmeal and transmitted virus by bite (no. feeding); Transmission rate[b]: % of orally exposed mosquitoes with a disseminated infection that took a second bloodmeal and transmitted virus by bite (no. feeding); Transmission rate[c]: % inoculated mosquitoes that transmitted virus by bite (no. feeding); Transmission rate[d]: % of all mosquitoes with a disseminated infection (inoculated orally exposed) that transmitted virus by bite (no. feeding); (··) Not defined; [] Lineage and inoculated virus titer (PFU/mL); n.t = not tested

^18–20 days after infective feed

^^ 14–15 days after infective feed.

tested, as reported in reference [22], and the lack of experimental infection at different temperatures would require further studies.

Interestingly, KOUTV viral dissemination and vertical transmission have been observed *in Ae. aegypti* mosquitoes in Senegal [12,54,55] while WNV vector competence experiments have never been conducted for this mosquito species in the African continent.

The ability of different mosquito species to transmit specific lineages of WNV highlights a direct correlation between vector competence and genetic variability [22].

Besides mosquitoes, ticks have been suggested as possible WNV hosts in Africa even though they are generally considered less competent as vectors compared to mosquitoes [6,7]. The role of ticks in WNV ecology and transmission is still an open question due to the little number of studies carried out so far on this topic [6,7,36,45,57–62]. Ticks are characterized by a protracted life cycle, holding the virus for a long time [61]. Furthermore, the transstadial maintenance of WNV in hard and soft ticks has been demonstrated [60,63,64].

Particular attention should be given to *Ornithodoros savignyi* and *Argas arboreus* ticks as potential vectors of WNV in the African continent. In Egypt, experimental infections of adult

soft tick *O. savignyi*, using a local strain of WNV (Ar-248), showed that the species got infected without being competent. However, after parenteral infection, *O. savignyi* could transmit the virus to infant mice and WNV isolation could be obtained from its coxal fluid [57].

*Argas arboreus* ticks are also shown to be WNV competent vectors and vertical and horizontal transmission has been observed for this species. The virus titer was detected to be 10^4 PFU/mL at 4 days post infection (pi), remaining constant at 10^3 PFU/mL from day 6 to day 50 pi. After 20 days from experimental infections, *A. arboreus* adults could transmit the virus to uninfected chickens. Furthermore, F1 *A. arboreus* larvae from WNV experimentally infected females could also transmit the virus to uninfected chickens. WNV was isolated from *A. arboreus* salivary glands, synganglia, and coxal fluids [59].

A large number of ticks are carried around the world by mammals and by birds during their migration paths [61]. Further studies are needed to assess the possible contribution of ticks on the maintenance, transmission, and spread of WNV over long distances and extensive periods of time. Considering that WNV competent hosts are characterized by relatively short viremic duration periods, the role of ticks as WNV vectors should be further evaluated especially for better explaining the biological mechanism which favors virus translocation from the African continent into the others [6,65]. Results of laboratory vector competence experiments and WNV isolation in hard and soft ticks are shown in Table 3.

## 3. West Nile virus epidemiology in African vertebrates

Several avian species are exposed to WNV infection as ascertained through sero-epidemiological studies or virus isolation but only for a subset of species their role as competent reservoir has been tested [7,8]. Susceptibility of birds to WNV infection is dependent on bird species other than to the viral strain involved [10]. In Europe and the United States, Passeriformes and Falconiformes appear to be highly susceptible to WNV-L1 and L2, showing neurological symptoms and variable mortality rates [7,10,67]. On the contrary, clinical signs have been rarely reported in birds in the African continent, with exception of one moribund pigeon affected by WNND in Egypt in 1953 [45].

However, the high viral circulation among birds in Africa is documented by seroprevalence studies carried out in Algeria, Egypt, Morocco, Tunisia, Southern Sudan, Senegal, Madagascar, and South Africa, as summarized in Fig 2 and Table 4 [34,41,45,56,68–76].

Among equids, symptomatic infections and fatalities have been reported in Morocco [WNV-L1] and South Africa (WNV-L2) [7,77–79]. In South Africa, WNV-L1 is rare and was detected only once in a lethal neurological case involving a mare and its aborted fetus during an eight years long observational study [79].

Seroprevalence studies carried out on horses and other equids and aimed at assessing the circulation of WNV infection have been carried out in Morocco, Tunisia, Egypt, Algeria, Nigeria, South Sudan, Democratic Republic of Congo, Chad, South Africa, Gabon, Côte d'Ivoire, Senegal, and Djibouti, as shown in Fig 2 and Table 5 [7,13,42,44,61,78–93].

Besides in birds and equids, WNV has been reported in other animal species [94,95]. Fig 2 and Table 6 summarize the recording of WNV exposure in other vertebrates in African countries although their role in the transmission cycle is not well understood yet [15,19,34,66,77,79,86,88,94–102].

## 4. West Nile virus distribution in humans

Several WNV outbreaks in humans were registered in the African continent starting from the 1950s [2,13,21,34,41,44,46,49,104–106]. Neurological cases and fatalities related to WNV-L2 were reported in South Africa (1976, 1980, 1984) while in the Mediterranean basin, hundreds

**Table 3. Viral isolation and laboratory vector competence experiments on hard and soft African ticks.**

| Year | Species | Country | Viral Isolation | Competence experiments | WNV Strain / Inoculated virus titer | Infection | Transmission | References |
|---|---|---|---|---|---|---|---|---|
| (··) | *Ornithodoros capensis* | Egypt | + | + | - | Yes | No | [7,36[c]] |
| (··) | *Ornithodoros erraticus* | Egypt | - | + | (··) | Yes | No | [36[c],45[c],60[a]] |
| 1950s | *Ornithodoros savignyi* | Egypt | - | + | Ar-248 strain | Yes | No | [45[c],60[a]] |
| 1950s | *Ornithodoros savignyi* | Egypt | + | + | (··) | Yes | Yes | [57[a]] |
| 1993, 2003 | *Argas hermanni* | Egypt, Senegal | + | + | $10^{5.5}$ TCID50/mL | Yes | No | [7,36[c],45[c],59[a],61] |
| 1993, 2003 | *Argas hermanni* | Egypt, Senegal | + | + | $10^{6.2}$ TCID50/mL | Yes | No | [7,36[c],45[c],59[a],61] |
| 1993, 2003 | *Argas persicurs* | Egypt, Senegal | - | + | $10^{5.5}$ TCID50/mL | Yes | No | [57[a],59[a],61] |
| 1993, 2003 | *Argas persicurs* | Egypt, Senegal | - | + | $10^{6.2}$ TCID50/mL | Yes | No | [57[a],59[a],61] |
| 1993, 2003 | *Argas arboreus* | Egypt | - | + | $10^{5.5}$ TCID50/mL | Yes | No | [57[a],59[a]] |
| 1993, 2003 | *Argas arboreus* | Egypt | - | + | $10^{6.2}$ TCID50/mL | Yes | Yes | [59[a]] |
| (··) | *Rhipicephalus turanicus* | Central African Republic | + | - | - | No | No | [36[c]],[66[a]] |
| (··) | *Rhipicephalus lunulatus* | Central African Republic | + | - | - | - | - | [66[a]],IPD[c] |
| (··) | *Rhipicephalus muhsamae* | Central African Republic, Senegal | + (WNV Central African Republic; KOUTV Senegal) | - | - | - | - | [12[a],66[a]],IPD[c] |
| (··) | *Amblyomma variegatum* | Central African Republic, Ivory Coast | + | - | - | No | No | [7[c],36[c],66[a]],IPD |
| 2010–2012 | *Rhipicephalus pulchellus* | Kenya | + | - | - | No | No | [62[a,b]] |
| (··) | *Hyalomma* | Africa | + | - | - | No | No | [7[c]] |
| 2010–2012 | *Amblyomma gemma* | Kenya | + | - | - | No | No | [7[c],62[a,b]] |
| (··) | *Dermacentor marginatus* | Africa | + | + | - | No | No | [7[c],62[a,b]] |
| (··) | *Hyalomma marginatum rufipes* | Senegal | + (KOUTV) | - | - | - | - | [12[a]],IPD[c] |
| (··) | *Ripicephalus guilhoni* | Senegal | + (KOUTV) | - | - | - | - | [12[a]],IPD[c] |

+ At least one study report with positive results found;—no available studies; (··) Not defined; IPD: Institut Pasteur de Dakar, Senegal, personal communication

[a] = "virus isolation on cell cultures / injection into suckling mice. Viruses detected by immunofluorescence assay using specific mouse immune ascitic fluids"

[b] = "RNA molecular detection"

[c] = "viral isolation, not specified if [a] and/or [b]"; reference numbers with no apical letters only refer to experimental infections.

of cases of encephalitis and deaths related to WNV-L1 (clade A) were registered in Tunisia between 1997 and 2018 (1997, 2003, 2007, 2010, 2011, 2012, 2016, 2018) [21,41,44,104,107]. In addition, WNV-L1 human infections were recorded for the first time in the 1994 and 1996 in Algeria and Morocco, respectively [44,82,105].

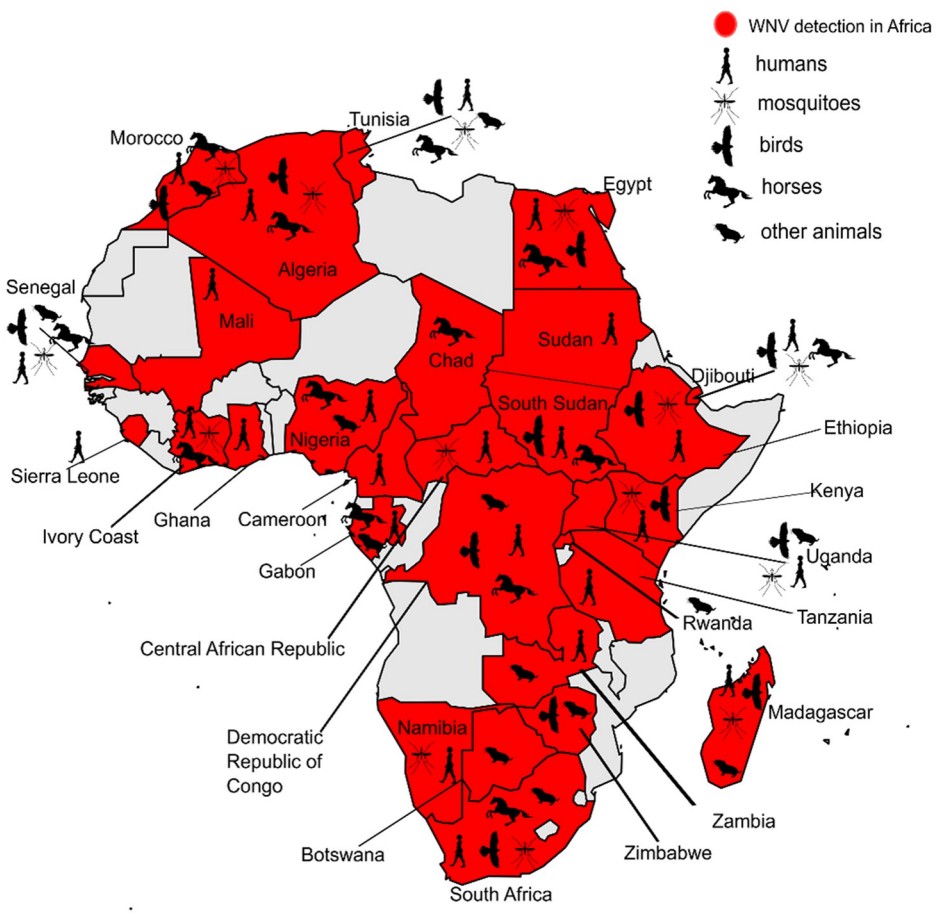

**Fig 2. Map of West Nile virus distribution in Africa based on sero-epidemiological surveys carried out on humans and animals, and viral isolation in mosquitoes.** Map was generated using publicly available shapefiles, https://smart.servier.com/category/general-items/world-maps/.

Real time PCR analysis confirmed the WNV circulation in the Central African Republic, Guinea, Ghana, Gabon, Nigeria, Senegal, and Sierra Leone between 1983 and 2020 [26,61,108–112] while serological surveys reported WNV circulation in humans in Algeria, Central African Republic, Democratic Republic of Congo, Egypt, Ethiopia, Gabon, Ghana, Kenya, Madagascar, Mali, Morocco, Mozambique, Namibia, Nigeria, Senegal, Sierra Leone, South Africa, South Sudan, Sudan, Tanzania, Tunisia, Uganda, and Zambia, as shown in Fig 2 and Table 7 [7,19,21,34,44,45,49,51,61,65,66,69,90,91,103–107,109–147]. On the contrary, no WNV antibodies were detected in sero-surveys conducted in Algeria [113], Burundi [114], Cameroon [115], Mozambique [131], and Nigeria [135].

**Table 4. West Nile virus records in avian species in Africa.**

| Species | Country | Antibodies detection | Viral detection | Case of illness | Experimental infection | References |
|---|---|---|---|---|---|---|
| *Acrocephalus gracilirostris* | South Africa | + | - | - | - | [34[d]] |
| *Anas erythrorhyncha* | South Africa | + | - | - | - | [34[d]] |
| *Anas platyrhynchos domesticus* | Egypt, Madagascar | + | - | - | - | [45[e],56[f]] |
| *Anas undulata* | South Africa | + | - | - | - | [34[d]] |
| *Anas platyrhynchos* | Tunisia | + | - | - | - | [41[f]] |
| *Anthus trivialis* | Senegal | + | - | - | - | [68[f]] |
| *Antichromus minutus* | Central African Republic | - | + | - | - | IPD[c] |
| *Bubulcus ibis* | Egypt, Southern Sudan, Nile Delta, South Africa | + | - | - | + | [34[d],45[e],70[g]] |
| *Cercotrichas podobe* | Senegal | + | - | - | - | [68[f]] |
| *Cercotrichas galactotes* | Senegal | + | - | - | - | [68[f]] |
| *Cettia cetti* | Morocco | + | - | - | - | [71[h]] |
| *Columba livia* | Egypt | + | + | + | + | [45[c,e],70[g]] |
| *Coracopsis vasa* | Madagascar | - | + | - | - | [72[c]] |
| *Corvus corone* sardonius | Egypt, Southern Sudan, Nile Delta | + | + | - | + | [45[c,e],70[g]] |
| *Anas platyrhynchos* | Tunisia | - | + | - | - | [41[b]] |
| *Egretta garzetta Linnaeus* | Madagascar | - | + | - | - | [72[c]] |
| *Estrilda melpoda* | Central African Republic | - | + | - | - | IPD[c] |
| *Euplectes orix* | South Africa | + | - | - | - | [34[d]] |
| *Falco tinnunculus* | Egypt, Southern Sudan, Nile Delta | + | - | - | + | [45[e],70[g]] |
| *Fulica cristata* | South Africa | + | - | - | + | [34[d]] |
| *Gallus gallus* | Egypt; Madagascar; Tunisia | + | + | - | + | [21[b],45[c,e],56[b,f],73[b]] |
| Goose [*Anatidae*] | Egypt, Madagascar | + | - | - | - | [45[e],56[f]] |
| Guinea fowl [*Numididae*] | Madagascar | + | - | - | - | [56[f]] |
| *Hippolais opaca* | Senegal | + | - | - | - | [68[f]] |
| *Hirundo rustica* | Zimbabwe | - | + | - | - | [74[b]] |
| *Jynx torquilla* | Senegal | + | - | - | - | [68[f]] |
| *Lanius senator* | Senegal | + | - | - | - | [68[f]] |
| *Meleagris* | Madagascar | + | - | - | - | [56[f]] |
| *Milvus migrans aegyptius* | Egypt | + | - | - | - | [45[e]] |
| *Oena Capensis* | Senegal | + | - | - | - | [68[f]] |
| *Passer domesticus* | Egypt, Southern Sudan, Nile Delta, Morocco, Algeria | + | - | - | + | [45[e],70[g],71[h],75[e]] |
| *Pelecanus onocrotalus* | Senegal | + | - | - | - | [76[f]] |
| *Ploceus cucullatus* | Senegal | + | - | - | - | [68[f]] |
| *Ploceus velatus* | Senegal, South Africa | + | - | - | + | [34,68[f]] |
| *Quelea quelea* | South Africa | + | - | - | + | [34[d]] |
| *Riparia paludicola* | Zimbabwe | - | + | - | - | [74[b]] |
| *Streptopelia vinacea* | Senegal | + | - | - | - | [68[f]] |
| *Streptopelia senegalensis* | Egypt, Southern Sudan, Nile Delta, Senegal, South Africa | + | - | - | + | [34[d],45[e],68[f],70[g]] |
| *Threskiornis aethiopicus* | South Africa | + | - | - | + | [34[d]] |
| *Tchagra australis* | Central African Republic | - | + | - | - | IPD[c] |
| *Turdus merula* | Morocco | + | - | - | - | [71[h]] |
| *Turdus philomelos* | Algeria | + | - | - | - | [75[e]] |

(*Continued*)

**Table 4.** (Continued)

| Species | Country | Antibodies detection | Viral detection | Case of illness | Experimental infection | References |
|---------|---------|----------------------|-----------------|-----------------|------------------------|------------|
| *Urocolius macrourus* | Senegal | + | - | - | - | [68[f]] |

+ At least one study report with positive results found;—no available studies; IPD: Institut Pasteur de Dakar, Senegal, personal communication

[a] = "virus isolation on cell cultures / injection into suckling mice. Viruses detected by immunofluorescence assay using specific mouse immune ascitic fluids"

[b] = "RNA molecular detection"

[c] = "viral isolation, not specified if [a] and/or [b];"

antibodies detection via

[d] = "hemagglutination-inhibition test (HIT)"

[e] = "serological surveys, type of antibodies detection tests non specified"

[f] = "epitope-blocking enzyme-linked immunosorbent assay (ELISA)"

[g] = "virus-neutralization test (VNT)"

[h] = "micro virus-neutralization test (micro-VNT)"

[i] = "immunoglobulin M (IgM)-specific ELISA"

[l] = "plaque reduction neutralization test (PRNT)"

[m] = "Flavivirus microsphere immunoassay (MIA)"

reference numbers with no apical letters refer to experimental infections or case of illness.

In more than 20 countries (Angola, Benin, Botswana, Burkina Faso, Chad, Congo Brazzaville, Eritrea, Equatorial Guinea, Guinea, Guinea-Bissau, Côte d'Ivoire, Lesotho, Liberia, Malawi, Libya, Mauritania, Niger, Rwanda, Somalia, Swaziland, The Gambia, Togo, Western Sahara, and Zimbabwe) no WNV seroprevalence studies on humans have been conducted so far. Therefore, the real disease burden for the African human population is currently largely underestimated.

## Discussion

This study, based on the analysis of 153 scientific papers published between 1940 and 2021, provides updated knowledge and data on the state of art on WNV investigation carried out in Africa, highlighting several knowledge gaps related to fundamental aspects of WNV ecology and epidemiology. They include the partial knowledge on the actual WNV distribution and genetic diversity, its ecology and transmission chains including the role of different arthropods and vertebrate species as competent reservoirs, and the real disease burden for humans and animals, therefore emphasizing the needs for further research studies to be addressed with high priority in this Continent.

Numerous reports highlight the circulation of WNV-L1, 2, and 8 in the African continent, where the most common ancestor originated between the 16th and 17th century, followed by the introduction of WNV-L1 into Europe and the Americas, and WNV-L2 into Europe [2,3]. The KOUV, highly virulent in mice and associated with a symptomatic infection in a clinical laboratory worker, is also occurring in the African continent. Its potential spread into Europe and the Americas, and a possible impact on human and animal health should be considered [2,12,22,150].

Nowadays, little is revealed about the spatio-temporal epidemiology of WNV, and genetic relationships between African, European, and American strains are mostly unknown [14]. All the strains circulating in America seem to be derived from a single introduction of WNV-L1, detected in North America in the 1999 [5]. Phylogenetic analysis support the hypothesis that this introduction was originated from Israel, as highlighted by genetic similarity of American

**Table 5. West Nile virus records in equids in Africa (period 1975–2015).**

| Country | Year of the study | Viral Isolation | Antibody detection | Seroprevalence rate | References |
|---|---|---|---|---|---|
| Algeria | 1975 | - | + | 96.6% | [44[e]] |
| Chad | 2003 | - | + | 97% | [78[f,n]] |
| Djibouti | 2004–2005 | - | + | 9% | [78[f,n]] |
| Democratic Republic of Congo | 2004 | - | + | 30% | [78[f,n]] |
| Egypt | 1963 | - | + | 54% | [80[g]] |
| Gabon | 2004 | - | + | 3% | [78[f]] |
| Ivory Coast | 2003 | - | + | 79% | [81[f]] |
| Ivory Coast | 2003–2004–2005 | - | + | 28% | [78[f]] |
| Morocco | 1996 (42 deaths) | + | - | - | [7[b]] |
| Morocco | After the epizootic of 1996 | - | + | 42–57% | [82[e]] |
| Morocco | 2003 (5 deaths) | + | - | - | [83[f]] |
| Morocco | 2010 (8 deaths) | + | - | - | [44[f]] |
| Morocco | 2011 | - | + | 31% | [84[f,g]] |
| Morocco | 2018 | - | + | 33.7% | [42[f,h,m]] |
| Nigeria | 2011–2012 | - | + | 90.3% | [85[f,i]] |
| Nigeria | 2014 | - | + | 11.5% | [86[g]] |
| Nigeria | 2014 | - | + | 8.5% (donkeys) | [86[g]] |
| Senegal | 2002–2003, Dakar | - | + | 92% | [78[f,l]] |
| Senegal | 2003, Ferlo area | - | + | 78.3% | [61[l]] |
| Senegal | 2005, Senegal river basin | - | + | 85% | [87[f,g]] |
| Senegal | 2014, North-west Senegal | - | + | 68.7% | [88[f]] |
| Senegal | 2014, Keur Momar Sarr | - | + | 86.2% | [88[f]] |
| South Africa | 2001 | - | + | 15% (foals) | [89[l]] |
| South Africa | 2001 | - | + | 11% (yearlings) | [89[l]] |
| South Africa | 2001 | - | + | 75% (dams) | [89[l]] |
| South Africa | 2007–2008 | - | + | 21.8% | [13[f,g]] |
| South Africa | 2008–2015 | - | + | 7.4% | [79[f,g]] |
| Tunisia | 1980 | - | + | 0.35% | [90[e]] |
| Tunisia | 2005 | - | + | 25% | [91[e]] |
| Tunisia | 2005 | - | + | 37% (donkeys & mules) | [91[e]] |
| Tunisia | 2007 | - | + | IgG 30% | [92[f]] |
| Tunisia | 2007 | - | + | IgM 0.78% | [92[f]] |
| Tunisia | 2008 | - | + | 27.1% | [93[f]] |

+ At least one study report with positive results found;—no available studies; () specified when not horses

[a] = "virus isolation on cell cultures / injection into suckling mice. Viruses detected by immunofluorescence assay using specific mouse immune ascitic fluids"

[b] = "RNA molecular detection"

[c] = "viral isolation, not specified if [a] and/or [b]"; antibodies detection via

[d] = "hemagglutination-inhibition test (HIT)"

[e] = "serological surveys, type of antibodies detection tests non specified"

[f] = "epitope-blocking enzyme-linked immunosorbent assay (ELISA)"

[g] = "virus-neutralization test (VNT)"

[h] = "micro virus-neutralization test (micro-VNT)"

[i] = "immunoglobulin M (IgM)-specific ELISA"

[l] = "plaque reduction neutralization test (PRNT)"

[m] = "flavivirus microsphere immunoassay (MIA)"

[n] = "immunoblotting method (WB)."

**Table 6. West Nile Virus records in other vertebrate species.**

| Species | Country | Antibodies detection | Viral detection | Case of illness | Experimental infection | References |
|---|---|---|---|---|---|---|
| African forest buffalo, *Syncerus caffer nanus* | Democratic Republic of Congo, Gabon, South Africa | + | + | - | - | [95[l],96[b]] |
| African elephant, *Loxodonta* | Zambia | + | - | - | - | [95[l]] |
| Calves, domestic bovid, *Bovidae* | South Africa | + | + | - | + | [34[q],66[q],96[b],98[d],103[n]] |
| Domestic dog, *Canis lupus familiaris* | South Africa, Senegal, Botswana | + | + | + | + | [7[p,d,g],19[c,r],34[q],88[f],94[r],96[b],97[d,g,o,p,q]] |
| Donkey, *Equus asinus* | Algeria, Senegal, Nigeria | + | - | - | - | [44[e],86[g],88[f]] |
| Fallow deer, *Dama dama* | South Africa | - | + | - | - | [96[b]] |
| Giraffe, *Giraffa* | South Africa | - | + | - | - | [96[b]] |
| Goat, *Capra aegagrus hircus* | Senegal, Nigeria, South Africa | + | + | - | + | [34[q],88[f],96[b],98[d]] |
| Humped camel, *Camelus bactrianus* | Morocco, Nigeria | + | - | - | - | [86[d],88[f], 98[d],99[f,g],103[n]] |
| Lemur, *Galago senegalensis* | Senegal | - | + | - | - | [66[c]] |
| Lion, *Panthera leo* | South Africa | - | + | - | - | [96[b]] |
| Livestock | South Africa | + | + | + | - | [79[b,f,g,r]] |
| Mountain gorillas, *Gorilla beringei beringei* | Uganda, Rwanda, DRC | + | - | - | - | [95[l]] |
| Oxen, *Bos* | Madagascar | + | - | - | - | [7[e]] |
| Pigs, *Sus* | South Africa | - | - | - | + | [7[q],66[q]] |
| Roan antelope, *Hippotragus equinus* | South Africa | - | + | - | - | [96[b]] |
| Small rodents, *Rodentia* | Nigeria, Morocco, Tunisia, South Africa | + | + | - | + | [77[c],94[d,n,q],100[p,q],102[d]] |
| Wild rodents | Senegal, Somalia, Central African Republic | - | + (KOUTV) | - | - | [12[c]] |
| Sheep, *Ovis aries* | South Africa, Nigeria | + | - | + | + | [66[o,p,q],98[d],101[r,q]] |
| Kuhl's pipistrelle, *Pipistrellus kuhli* | Tunisia | + | - | - | - | [102[d]] |
| Wildlife | South Africa | + | + | + | - | [79[b,f,g]] |
| White rhinoceros *Ceratotherium simum* | South Africa | + | | | | [96[f]] |

+ At least one study report with positive results found;—no available studies

[a] = "virus isolation on cell cultures / injection into suckling mice. Viruses detected by immunofluorescence assay using specific mouse immune ascitic fluids"

[b] = "RNA molecular detection"

[c] = "viral isolation, not specified if [a] and/or [b]"; antibodies detection via

[d] = "hemagglutination-inhibition test (HIT)"

[e] = "serological surveys, type of antibodies detection tests non specified"

[f] = "epitope-blocking enzyme-linked immunosorbent assay (ELISA)"

[g] = "virus-neutralization test (VNT)"

[h] = "micro virus-neutralization test (micro-VNT)"

[i] = "immunoglobulin M (IgM)-specific ELISA"

[l] = "plaque reduction neutralization test (PRNT)"

[m] = "flavivirus microsphere immunoassay (MIA)"

[n] = "complement fixation test (CFT)"; "Experimental infection" means [o] = "disease"

[p] = "antibodies" or [q] = "attempt without any clinical signs/development of viraemia"

[r] = "case of illness".

**Table 7. West Nile virus seroprevalence studies carried out in humans between 1950 and 2019.**

| Country | Year of the study | Seroprevalence rate | References |
|---|---|---|---|
| Algeria | 1965 | 0% | [44[e],113[e]] |
| Algeria | 1973, 1975 | 14.6% | [44[e],113[e]] |
| Algeria | 1973, 1975 | 58.3% | [44[e],113[e]] |
| Algeria | 1973, 1975 | 3.5% | [44[e],113[e]] |
| Algeria | 1976 | 37.5% | [44[e],113[e]] |
| Algeria | 1976 | 19% | [44[e],113[e]] |
| Algeria | 1994 | 83.3% | [44[e]] |
| Burundi | 1980–1982 | 0% | [114[e]] |
| Cameroon | 1990 | 0% | [115[f,o]] |
| Cameroon | 2000–2003 | 6.6% | [116[l]] |
| Central African Republic | 1964 | High arbovirus circulation (··) | [117[e]] |
| Central African Republic | 1975–1976 | 2.3% | [117[e]] |
| Central African Republic | 1979 | WNV-positive results (··) | [118[d,n]] |
| Democratic Republic of Congo | 1998 | 66% | [119[i]] |
| Egypt | 1950 | 70% | [105[g,n]] |
| Egypt | 1951–1954 | 61% (44% < 15 years old; 72% > 15 years old) | [105[g,n]] |
| Egypt | 1952 | (··) | [148[e]] |
| Egypt | 1999–2002 | 35% Upper Egypt | [106[f,l]] |
| Egypt | 1999–2002 | 27% Middle Egypt | [106[f,l]] |
| Egypt | 1999–2002 | 14% Lower Egypt | [106[f,l]] |
| Egypt | 1999–2002 | 1% North Sinai | [106[f,l]] |
| Egypt | 1999–2002 | 7% South Sinai | [106[f,l]] |
| Ethiopia | 1959–1962 | (··) | [149[e]] |
| Gabon | 1975 | (··) | [121[f]] |
| Gabon | 1975 | KOUTV (··) | [12[e]] |
| Gabon | 21st century | 27.2% | [121[f]] |
| Ghana | 2008 | Children: 1.4% IgM, 4.8% IgG; Adults: 27.9% | [109[f]] |
| Kenya | 1959–1962 | (··) | [149[e]] |
| Kenya | 1966–1968 | 3.2% Central Nyanza | [122[d]] |
| Kenya | 1966–1968 | 13.8% Kitui District | [122[d]] |
| Kenya | 1966–1968 | 65.3% Malindi district | [122[d]] |
| Kenya | 1987 | 0.9% | [123[P]] |
| Kenya | 2009–2012 | 12.4% | [124[f],125[f,l]] |
| Kenya | 2016–2017 | 10.2% Turkana | [126[l]] |
| Madagascar | Since 1975 | (··) | [51[d]] |
| Madagascar | 1996 | 2.1% | [127[e]] |
| Madagascar | 1999 | 10.6% | [127[e]] |
| Madagascar | 2011 | IgM antibodies | [128[l,f]] |
| Mali | 2009–20013 | 0.27% IgM | [129[f]] |
| Mali | 2009–20013 | 39.1% IgG | [129[f]] |
| Morocco | 2011 | 11.8% (4.5% Meknes; 12% Rabat; 18.8% Kenitra) | [130[l]] |
| Morocco | 2019 | 4.39% positive to flaviviruses (75% of which confirmed WNV + by VNT) | [69[f,g]] |
| Mozambique | 2012–2013 | 0% | [131[f,o]] |
| Mozambique | (··) | (··) | [34[e]] |
| Namibia | 1983 | % | [132[e]] |
| Nigeria | 1970s | 28% | [133[d]] |

*(Continued)*

**Table 7.** (Continued)

| Country | Year of the study | Seroprevalence rate | References |
|---|---|---|---|
| Nigeria | 1990s | 65% | [133[d]] |
| Nigeria | 2008 | 25% | [107[f]] |
| Nigeria | 2011–2012 | 73.2% | [134[f,l]] |
| Nigeria | 2013 | 0% IgM | [135[i]] |
| Nigeria | 2016 | 7.5% IgM | [136[i]] |
| Nigeria | 21st century | 1.2% IgM; 80.16% IgG | [7[f,l]] |
| Nigeria | 21st century | 40% | [7[d,f]] |
| Senegal | 1972–1975 | (··) | [120[d,]] |
| Senegal | 1988–1990 | IgM < 15 years old (4.6% out of 456 and 3.5% out of 396 children tested) | [137[f]] |
| Senegal | 1989 | 80% (5–15 years old; 45% < 5 years old; 98% > 15 years old) | [61[l],66[f]] |
| Senegal | 1991 | 22.7% of adults; 18% < than 15 years old | [66[f]] |
| Sierra Leone | 2006–2008 | IgG in 50% of patients presenting severe symptoms, IgM 1/4 of them | [111[f]] |
| Sierra Leone | 2006–2008 | 1.2% IgM | [138[i]] |
| South Africa | 1950 | 2.6% | [139[e]] |
| South Africa | 1960s | 4.7% | [140[d]] |
| South Africa | (··) | 1% | [141[d,g]] |
| South Africa | 1962–1964 | 10.22% | [140[d]] |
| South Africa | 1970s | 7% Central Highveld Region; 17% Karoo; 2% Kwazulu Natal | [7[f,o]] |
| South Africa | 1974 | 55% - 85% | [19[e],49[e]] |
| South Africa | 2009 | 17.47% | [142[f,g]] |
| South Africa | 2017 | woman (IgM positive—2 weeks later IgG positive), man (IgM and IgG at 5 days after the beginning of the symptoms) | [143[f]] |
| South Sudan | 1951–1954 | 40% | [45[e],105[g,n]] |
| Sudan | After the epidemics of 1998 | 59% IgG antibodies | [144[p]] |
| Tanzania | 1971 | 17.4% | [145[g]] |
| Tunisia | 1968 | 1.80% | [90[d,f]] |
| Tunisia | 1970s | 4.7% (3.8% Djerba region; 7.8% Tunis; 7% Gabes; 9% other Tunisian regions) | [91[g,o]] |
| Tunisia | 1997 | 86%, including 5 fatalities | [7[f,i],105[n,n],107[f]] |
| Tunisia | 1997 | 9 IgM positive results | [7[f,i],105[n,n],107[f]] |
| Tunisia | 2007 | 12·5% (27.7% Kerouan, 7.5% Sfax, 0.7% Bizerte) | [91[g,o],104[f,l]] |
| Tunisia | 2018 | 24% | [21[i]] |
| Uganda | 1984 | 16% of anti-flavivirus antibodies (probably due to WNV) | [146[d]] |
| Zambia | 2010 | 10.3% | [147[f]] |

(··) Not defined; antibodies detection via

[d] = "hemagglutination-inhibition test (HIT)"

[e] = "serological surveys, type of antibodies detection tests non specified"

[f] = "epitope-blocking enzyme-linked immunosorbent assay (ELISA)"

[g] = "virus-neutralization test (VNT)"

[h] = "micro virus-neutralization test (micro-VNT)"

[i] = "immunoglobulin M (IgM)-specific ELISA"

[l] = "plaque reduction neutralization test (PRNT)"

[m] = "flavivirus microsphere immunoassay (MIA)"

[n] = "complement fixation test (CFT)"

[o] = "indirect immunofluorescent assays (IFA)"

[p] = "enzyme immunoassay (EIA)"

strains with certain Israeli strains [14]. These strains, grouped in the Israeli-American cluster, are characterized by wild bird mortality and fatal encephalitis in humans and horses [5,8]. A close similarity has been observed between the Israeli-American strains (1998–2000) and the *PaH001* Tunisian strain of 1997, supporting the hypothesis of a possible flow of WNV between Africa and the New World via the middle East [14]. This hypothesis is corroborated by i) the enormous avian biodiversity of Tunisia, considered an important flyway for birds migrating from Africa to northern countries [73]; ii) the circulation of WNV in *Cx. pipiens* competent mosquitoes, birds, horses, and humans in the country [41,91] and iii) the WNV-L1 meningo-encephalitis outbreak, characterized by 173 human cases and 8 deaths, occurred in Tunisia in 1997, one and three years before the first detection of WNV in Israel and United States, respectively [44,104,107].

In Europe, WNV-L1, first detected in the 1960s, re-emerged in the Continent in the 1990s [5]. Since then, WNV-L1 strains, belonging to the Western-Mediterranean clade (Morocco 1996, Italy 1998, France 2000) and to the Eastern-European clade (Romania 1996, Russia 1999), caused numerous outbreaks in European countries and North Africa [3]. These strains were characterized by moderate pathogenicity for horses and humans and limited or no pathogenicity for birds [5]. A possible European WNV-L1 introduction from Morocco is suggested: the closest ancestor of the European strains may be a Moroccan strain which appears to be genetically related to French and Italian isolates (France: 2000, 2006; Italy: 1998, 2008) [3,6,20,83].

WNV-L2, for long time believed to be restricted to Sub-Saharan Africa and considered not pathogenic, is nowadays endemic and the most prevalent in several African and European countries, provoking clinical symptoms (main neurologic signs of infection include ataxia, weakness, recumbence, seizures and muscle fasciculation) among horses in South Africa [7,79], and pathogenesis among horses, humans and birds in Europe [3]. The exact origin of WNV-L2 strains and the following route of introduction into Europe is not clear [3,6]. In Europe, WNV-L2 was reported for the first time in Hungary in 2004 [3,15,151]. Since 2008, an increase in its transmission has been observed in many European countries (Austria, Greece, Italy, Serbia, Bulgaria, Romania, Spain, and Germany) [3]. Interestingly, in Africa WNV-L2 has been reported in Sub-Saharan African countries (Botswana, Central African Republic, Congo, Djibouti, Madagascar, Mozambique, Namibia, Senegal, South Africa, Tanzania, and Uganda) but never in Northern African countries, suggesting a possible flow between Sub-Saharan Africa and Europe, via the Nile Delta and the Mediterranean Sea through migratory birds.

These reports evidence the active circulation of WNV-L1 and L2 in Africa and the possible viral spread into Europe and the Americas, further emphasizing the need of a coordinated surveillance in Africa and Europe and the necessity of intensifying WNV research.

In Africa, WNV isolation on cell cultures and RNA molecular detection have been obtained from 46 mosquito species, as shown in Table 1. Many of these mosquito species have not been tested for their competence for WNV (Table 2) and therefore they should be the future subject for WNV laboratory vector competence experiments.

Vector competence experiments conducted in Madagascar, Algeria, Morocco, Tunisia, Senegal, Kenya, and South Africa highlight the vector competence of *Cx. pipiens*, *Cx. quinquefasciatus*, *Cx. univittatus*, *Cx. theileri*, *Cx. vansomereni*, and *Cx. neavei* mosquitoes in Africa [22,29,33–35,50]. In particular, our review highlights the high competence of *Cx. pipiens* and *Cx. univittatus* mosquitoes while transmission rates in *Cx. neavei* and *Cx. quinquefasciatus* seem to be lower, due to estimated short longevity and long EIP [22]. Further research is needed to confirm these findings and assess their impact on WNV circulation in Africa.

Particular attention should be paid towards *Ae. aegypti* mosquitoes in Africa. Although WNV has been isolated in wild specimens belonging to this species [51], which was also found to be capable of KOUTV transmission in Senegal [22,54,55]. *aegypti* African populations have never been tested for WNV vectorial competence so far. However, laboratory studies conducted in the USA demonstrated the vector competence of local *Ae. aegypti* strains for WNV, although the species was found to be less efficient than *Cx. pipiens* [152].

Interestingly, unusual cases of WNV transmission have been highlighted in *O. savignyi* and *A. arboreus* ticks in Egypt [57,59]. In particular, *O. savignyi* has been proven to be competent after parenteral infection while vertical and horizontal transmission have been shown for *A. arboreus* tick species. The role of ticks in transmission and maintenance of WNV should therefore be further explored.

Several WNV seroprevalence studies carried out on humans and animals have been reported, providing evidence of an intense WNV circulation in the Continent. In particular, as illustrated by Fig 2 and reported by the studies of Tables 4, 5, 6 and 7, seropositivity in humans and animals to WNV has been reported in Morocco, Tunisia, Algeria, Egypt, Mali, Senegal, Sierra Leone, Ivory Coast, Gabon, Ghana, Cameroon, Nigeria, Chad, Sudan, South Sudan, Djibouti, Ethiopia, Kenya, Tanzania, Central African Republic, Rwanda, Uganda, Zambia, Namibia, Botswana, South Africa, and Madagascar. Diverse serological methods have been used in different countries, ranging from HIT, ELISA, VNT, micro-VNT, PRNT, MIA, CFT, IFA, WB, and EIA. For 19 out of 99 serological studies (19.19%) [41,56,57,66,68,76,88,92,93, 96,107,109,111,121,124,129,137,143,147] only ELISA test was carried out without a specific WNV neutralization test, and therefore potential of cross-reactivity with closely related pathogens, such as Usutu virus, St. Louis encephalitis virus, or Japanese Encephalitis virus cannot be excluded [91].

Finally, detailed information on the serological tests carried out are not available for 18 of these studies (18.18%) [12,19,34,44,45,49,75,82,90,91,113,114,117,127,132,139,148,149]. It would be extremely important, in the next future surveys, to carry out WNV confirmation tests with standard methods such as micro-VNT or PRNT, in order to obtain specific serological responses, particularly in areas characterized by circulation of several Flaviviruses [121]. Unfortunately, many times the low volume of collected samples does not allow to perform further laboratory tests. Despite these limitations, these serologic findings associated with viral isolation through cell culture and RNA molecular detection in mosquitoes, few ticks, birds, horses, humans, and other mammals indicate that WNV is actively circulating in many areas of the African continent.

There are no studies available on WNV for Angola, Benin, Burkina Faso, Congo Brazzaville, Eritrea, Equatorial Guinea, Guinea, Guinea-Bissau, Lesotho, Liberia, Malawi, Libya, Mauritania, Niger, Somalia, Swaziland, The Gambia, Togo, and Western Sahara, highlighting the lack of information for several African countries. For this reason, the real burden of WNV infections in Africa may differ from what is currently reported by the published literature. However, despite the limitations due to a lack of observational and clinical data for a number of countries, the available information summarized in this review contribute to fill an existent knowledge gap on the phylogeography, ecology and epidemiology of WNV in Africa.

## Conclusion

Since its discovery in 1937 in Uganda, WNV has spread beyond its original ecological niches, becoming one of the most widespread viruses in the world and a serious public and veterinary health concern. Given the global burden of the virus, a deepened knowledge of its phylogeny, epidemiology and circulation in the African continent acquires increasing importance to

predict, identify and control WNV, but also other viruses as KOUV future epidemics. The actual epidemiological situation in most countries of the African continent is unknown, due to: i) the non-specificity of clinical signs of WNV infection with respect to other arboviruses, very often indistinguishable from each other and from other tropical diseases such malaria, typhoid fever or undifferentiated febrile illness [7,126]; ii) the poor information management and sharing of public health data system of most African countries [153]; iii) the possible bias obtained by serological analysis due to the utilization of non-specific WNV neutralization assays, where the risk of cross-reactivity among closely related pathogens cannot be excluded, and iv) the restricted availability of diagnostic capacity, lack of awareness and inconsistent surveillance, with most investigations performed only during outbreak periods [7,78,125]. All these factors make quantification of the yearly WNV circulation at continental level difficult, with only limited accurate data available in North and Sub-Saharan Africa.

New surveillance strategies for preventing and control WNV in Africa need to be implemented not only to better assess the current health impact but also to prevent and control future outbreaks, and therefore limit disease transmission. National and district levels should cooperate with Ministries of Health, and other international partners such as the WHO and European public and veterinary health bodies, to implement national surveillance programs in African countries and to coordinate surveillance actions between Africa and Europe. These programs core objectives should i) minimize the suffering and damage, ii) prevent national and international spread, and iii) contain outbreaks through strong surveillance for early detection and rapid response, as done before for other viruses, as the Yellow Fever and Polio-myelitis viruses.

The ability to accurately identify pathogens in a timely and accurate way has grown in the last few decades, and it has become increasingly clear that the implementation of a collaborative international and multi-disciplinary One Health action, based on the analyses of the interconnection among environmental, humans' and animals' health factors in different Continents, will be crucial to allow a more accurate risk assessment and thus an early response to West Nile virus but also to other emerging zoonotic pathogens by public human and veterinary health actors.

## Supporting information

**S1 Fig. Flow chart of the study selection process.**
(TIF)

## Acknowledgments

The contents of this publication are the sole responsibility of the authors and do not necessarily reflect the views of the European Commission.

## Author Contributions

**Conceptualization:** Giulia Mencattelli, Roberto Rosà, Giovanni Marini, Mawlouth Diallo, Giovanni Savini, Annapaola Rizzoli.

**Data curation:** Giulia Mencattelli.

**Formal analysis:** Giulia Mencattelli.

**Funding acquisition:** Giovanni Savini, Annapaola Rizzoli.

**Methodology:** Giulia Mencattelli, Roberto Rosà, Giovanni Marini.

**Project administration:** Roberto Rosà, Giovanni Savini, Annapaola Rizzoli.

**Resources:** Marie Henriette Dior Ndione, Giovanni Marini, Moussa Moise Diagne, Mawlouth Diallo, Giovanni Savini.

**Supervision:** Roberto Rosà, Giovanni Marini, Annapaola Rizzoli.

**Validation:** Marie Henriette Dior Ndione, Roberto Rosà, Giovanni Marini, Cheikh Tidiane Diagne, Moussa Moise Diagne, Gamou Fall, Mawlouth Diallo, Oumar Faye, Giovanni Savini, Annapaola Rizzoli.

**Visualization:** Marie Henriette Dior Ndione, Roberto Rosà, Giovanni Marini, Ousmane Faye, Mawlouth Diallo, Giovanni Savini, Annapaola Rizzoli.

**Writing – original draft:** Giulia Mencattelli.

**Writing – review & editing:** Giulia Mencattelli, Marie Henriette Dior Ndione, Roberto Rosà, Giovanni Marini, Cheikh Tidiane Diagne, Moussa Moise Diagne, Gamou Fall, Mawlouth Diallo, Oumar Faye, Giovanni Savini, Annapaola Rizzoli.

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
