## [Decision Letter · Decision Letter 0]

24 Aug 2021

Dear Dr Rizzoli,

Thank you very much for submitting your manuscript "Epidemiology of West Nile virus in Africa: an underestimated threat" for consideration at PLOS Neglected Tropical Diseases. As with all papers reviewed by the journal, your manuscript was reviewed by members of the editorial board and by several independent reviewers. In light of the reviews (below this email), we would like to invite the resubmission of a significantly-revised version that takes into account the reviewers' comments. 

We cannot make any decision about publication until we have seen the revised manuscript and your response to the reviewers' comments. Your revised manuscript is also likely to be sent to reviewers for further evaluation.

Sincerely,

Hans-Peter Fuehrer

Deputy Editor

Elvina Viennet

Deputy Editor

Reviewer's Responses to Questions

**Key Review Criteria Required for Acceptance?**

**Methods**

-Are the objectives of the study clearly articulated with a clear testable hypothesis stated?

-Is the study design appropriate to address the stated objectives?

-Is the population clearly described and appropriate for the hypothesis being tested?

-Is the sample size sufficient to ensure adequate power to address the hypothesis being tested?

-Were correct statistical analysis used to support conclusions?

-Are there concerns about ethical or regulatory requirements being met?

Reviewer #1: Methods used for particular serological surveys must be notified.

Reviewer #2: The article is a literature review that follows PRISMA and QUORUM criteria (referenced). The approach is appropriate and the authors have identified a considerable number of articles that are relevant to the study.

The authors have made contact with experts (n = 29) to obtain further information although what this consists of is not explained or the identity of these individuals.

No statistical analysis was conducted on the review.

There are not ethical or regulatory concerns on this manuscript.

Reviewer #3: This is a review article. The methods of publication selection is clearly described.

**Results**

-Does the analysis presented match the analysis plan?

-Are the results clearly and completely presented?

-Are the figures (Tables, Images) of sufficient quality for clarity?

Reviewer #1: Serological methods used for serosurveys data need to be notified, and possible cross-reactions with other African flaviviruses described, if possible. Modification is also required for "virology" data (virus isolation or RNA molecular detection).

Reviewer #2: The analysis does reflect the aims of the review. The authors have compiled a considerable volume of information concerning the epidemiology of West Nile virus in Africa presented in a series of seven tables and cite 150 references. However, a better description of West Nile virus phylogeny should be presented in lines 81/82 to explain what lineages 3,4,5,6 & 7 are, their relationship to the main lineages 1 and 2 and why they have not been detected in Africa, where presumably WNV evolved.

The authors should clarify on lines 135 to 136 that only lineage 1 emerged in the Americas.

The assertion on lines 143 to 144 that because an isolate from Tunisia roots the LI isolates from North America suggests "a possible flow between Tunisia and the Americas" ignores the presence of an isolate within that clade from Israel (1998). This needs to be revised to suggest that phylogeny suggest that viruses circulating in the Middle East / North Africa are related to those in North America.

The lines 198 to 202 are not explained very well and do not critically assess the papers they cite. They need to define EIP, state the small number of mosquitoes that were infected in reference 21 and if these species are poorly competent for WNV, how does the virus persist?

The discussion on line 218 on tick transmission of WNV is highly speculative.

A general comment is that the authors have documented a large volume of data but this is not matched by the discussion in the text, which could be improved.

Reviewer #3: The results extracted from the literature has been summarized very well and are presented in a concise way.

**Conclusions**

-Are the conclusions supported by the data presented?

-Are the limitations of analysis clearly described?

-Do the authors discuss how these data can be helpful to advance our understanding of the topic under study?

-Is public health relevance addressed?

Reviewer #1: Conclusions are well formulated, except for serosurveys.

Reviewer #2: This subject is neglected and more investigations of West Nile virus epidemiology are needed. However, as the authors state, biological science in many African countries (of which there are 54 recognized by the United Nations) are under-resourced and diagnostics for the virus limited. It then seems odd to then advocate for a widespread and long-term WNV surveillance system for the continent when there is no in-country activity. Who would provide the coordination to acheive this and who would fund it?

The authors mention climate change on line 363, seems a bit late to include this in the final sentances.

Reviewer #3: The conclusions are supported by the various publications, which have been thoroughly analyzed by the authors.

**Editorial and Data Presentation Modifications?**

Reviewer #1: Remarks and suggestions:

Line 79: ... such as birds, corvids and raptors... corvids and raptors are birds.

Line 189: Ae. aegypti is really a recognized vector of WNV? 

Line 204 (Tab.1): “Isolation” here means the virus isolation or also virus RNA molecular detection? Be more specific and differentiate, please.

Line 207 (Tab.2): Infection rate 14·28% means 14.28% (decimal point) or 14-28% ? (and several similar cases). “Virus titer”: inoculated or recovered titer? Explain, please.

Line 222 (Tab.3): The (unusual) cases of WNV transmission by Ornithodoros savignyi and Argas arboreus ticks need to be decribed (commented) in detail.

Line 237 (Tab.4): “Antibodies detection”: describe the serological test used (whether VNT, PRNT, HIT, CFT or ELISA), and a differentiation from other African flaviviruses (if done). This is (serology data) a weak point of the whole synopsis; flaviviruses are namely serologically readily cross-reactive and some serological results without a differentiation among related flaviviruses could thus be erroneous.

Line 237 (Tab.4): “Viral detection” means the virus isolation or viral RNA detection? Be more specific and differentiate, please.

Line 249 (Tab.5): “Viral isolation” means the virus isolation or viral RNA detection? Be more specific and differentiate, please.

Line 249 (Tab.5): “Antibody detection” and “Seroprevalence rate” describe the serological tests used, and differentiation from other African flaviviruses, if done. 

Line 255 (Tab.6): “Antibody detection” and “Viral detection” – see comments above. 

Line 255 (Tab.6): “Experimental infection” + means disease, antibodies or only attempt without any signs?

Line 262: outbreaks

Line 267: possibly a redundant sentence.

Line 269: Tunisia is already mentioned on the line 264.

Line 276 (Tab.7): “Seroprevalence rate” – see above.

Line 318: why you do not quote original paper of Hungarian record on WNV-2 ?

Line 326-331: are all these “isolations” from so many mosquito species really WNV isolations, or also WNV RNA molecular detections? Check it, please, and note.

Reviewer #2: The general format of the paper is satisfactory although a few minor changes are needed.

Lines 81 to 98, its unclear why there are six 1/2 sentance paragraphs. These need to be re-written into larger sections.

The final words of the introduction are a word for word repeat of the abstract, revise.

Line 326, why are all these mosquito species listed, surely just refer to one of the tables?

Reviewer #3: The only minor comment I have is:

Line 81: "WNV currently includes eight phylogenetic lineages [11]. Among all these observed eight lineages, only ...". Please change this sentence to: "WNV currently includes UP TO NINE phylogenetic lineages [11]. Among THESE LINEAGES, only ...". Instead of ref. [11] I suggest to use here ref. [29] (Pachler et al.).

**Summary and General Comments**

Reviewer #1: Presented serosurveys data in this synopsis are the weak point of the manuscript; flaviviruses are namely serologically readily cross-reactive and some "positive" serological results without a differentiation among related flaviviruses could thus be erroneous (in certain cases it might be unclear whether specific anribodies against West Nile virus were really detected).

Reviewer #2: The manuscript "Epidemiology of West Nile virus in Africa: an underestimated threat" provides a comprehensive review of West Nile virus epidemiology in Africa. This is a neglected subject and worth discussing. As described in previous section, the manuscript could be improved and the authors should be more critical of the subject and in their discussion in the text. The conclusions should also be of more practical benefit.

Reviewer #3: This article is a nice review of the West Nile virus epidemiology in Africa. It summarizes in a clear and concise way the topic.

PLOS authors have the option to publish the peer review history of their article (what does this mean?). If published, this will include your full peer review and any attached files.

Reviewer #1: No

Reviewer #2: No

Reviewer #3: No
---

## [Decision Letter · Decision Letter 1]

3 Dec 2021

Dear Dr Rizzoli,

Thank you very much for submitting your manuscript "Epidemiology of West Nile virus in Africa: an underestimated threat" for consideration at PLOS Neglected Tropical Diseases. As with all papers reviewed by the journal, your manuscript was reviewed by members of the editorial board and by several independent reviewers. The reviewers appreciated the attention to an important topic. Based on the reviews, we are likely to accept this manuscript for publication, providing that you modify the manuscript according to the review recommendations. 

Sincerely,

Hans-Peter Fuehrer

Deputy Editor

Elvina Viennet

Deputy Editor

Reviewer's Responses to Questions

**Key Review Criteria Required for Acceptance?**

**Methods**

-Are the objectives of the study clearly articulated with a clear testable hypothesis stated?

-Is the study design appropriate to address the stated objectives?

-Is the population clearly described and appropriate for the hypothesis being tested?

-Is the sample size sufficient to ensure adequate power to address the hypothesis being tested?

-Were correct statistical analysis used to support conclusions?

-Are there concerns about ethical or regulatory requirements being met?

Reviewer #2: OK

Reviewer #3: The methods are clearly presented in this review article.

**Results**

-Does the analysis presented match the analysis plan?

-Are the results clearly and completely presented?

-Are the figures (Tables, Images) of sufficient quality for clarity?

Reviewer #2: OK

Reviewer #3: The metaanalysis of the results was perfectly carried out.

**Conclusions**

-Are the conclusions supported by the data presented?

-Are the limitations of analysis clearly described?

-Do the authors discuss how these data can be helpful to advance our understanding of the topic under study?

-Is public health relevance addressed?

Reviewer #2: OK

Reviewer #3: The conclusions are presented well as a summary of the publications.

**Editorial and Data Presentation Modifications?**

Reviewer #2: OK

Reviewer #3: not needed.

**Summary and General Comments**

Reviewer #2: The authors have adequately addressed all the reviewers comments. The only issue now is that the text in places is unclear and could be improved throughout. The list below is only what this reviewer spotted but I would recommend that the authors check the manuscript again for corrections:

Line 84, no need to capitalise public health.

Line 86, correct to North America

Line 89, ".. one of the main lineages responsible for WNV..."

Line 91, South Moravia is a part of the Czech Republic, rephrase the sentence.

Line 98, continents

Line 134, correct to " We identified 408 articles. After duplicates were removed, the remaining 395 records were screened by title, abstract and full text."

Line 140, "WNV is a biologically diverse virus..."

Line 142, "...to be between the..."

Line 161, "WNV-L2 was considered for a long time to be less pathogenic..."

Line 170, "..only cluster 3 occurs in Africa.."

Line 333, Section 4 deals with the distribution of WNV in Africa rather than epidemiology, suggest change the title.

Line 383, "..1960s, re-emerged.."

Line 390, "..countries, causing disease among horses..." (horse can't describe symptoms?)

Line 402, Still not clear why all these species are listed. Suggest replace with "Many of these mosquito species have not been tested for their competence for WNV...."

Line 422, what is "paternal infection"?

Line 433, overuse of "the" delete them all and it reads better.

Line 436, ".. such as micro-VNT.."

Line 437, "Unfortunately, many times the nature, the remaining volume, and the storage procedures of the samples, do not allow to do so." No idea what this means, suggest revision.

Reviewer #3: The authors did a great job with this review and they included in their revised manuscript the suggestions of the reviewers. Congratulations to the authors to a very nice piece of work!

PLOS authors have the option to publish the peer review history of their article (what does this mean?). If published, this will include your full peer review and any attached files.

Reviewer #2: No

Reviewer #3: No

Figure Files:

Data Requirements:

Reproducibility:

References

---

## [Editor Report · Decision Letter 2]

9 Dec 2021

Dear Dr Rizzoli,

We are pleased to inform you that your manuscript 'Epidemiology of West Nile virus in Africa: an underestimated threat' has been provisionally accepted for publication in PLOS Neglected Tropical Diseases.

Best regards,

Hans-Peter Fuehrer

Deputy Editor

Elvina Viennet

Deputy Editor

Editor:

Please make following changes when getting the proofs: sp. and spp. not in italics

---

## [Editor Report · Acceptance letter]

5 Jan 2022

Dear Dr Rizzoli,

We are delighted to inform you that your manuscript, "Epidemiology of West Nile virus in Africa: an underestimated threat," has been formally accepted for publication in PLOS Neglected Tropical Diseases.

Best regards,

Shaden Kamhawi

co-Editor-in-Chief

Paul Brindley

co-Editor-in-Chief
